# Male survivorship and the evolution of eusociality in partially bivoltine sweat bees

**Jodie Gruber** **\*, Jeremy Field**

College of Life and Environmental Science, University of Exeter, Penryn Campus, Cornwall, United Kingdom

* j.gruber@exeter.ac.uk

## Abstract

Eusociality, where workers typically forfeit their own reproduction to assist their mothers in raising siblings, is a fundamental paradox in evolutionary biology. By sacrificing personal reproduction, helpers pay a significant cost, which must be outweighed by indirect fitness benefits of helping to raise siblings. In 1983, Jon Seger developed a model showing how in the haplodiploid Hymenoptera (ants, wasps and bees), a partially bivoltine life cycle with alternating sex ratios may have promoted the evolution of eusociality. Seger predicted that eusociality would be more likely to evolve in hymenopterans where a foundress produces a male-biased first brood sex ratio and a female-biased second brood. This allows first brood females to capitalize on super-sister relatedness through helping to produce the female-biased second brood. In Seger's model, the key factor driving alternating sex ratios was that first brood males survive to mate with females of both the second and the first brood, reducing the reproductive value of second brood males. Despite being potentially critical in the evolution of eusociality, however, male survivorship has received little empirical attention. Here, we tested whether first brood males survive across broods in the facultatively eusocial sweat bee *Halictus rubicundus*. We obtained high estimates of survival and, while recapture rates were low, at least 10% of first brood males survived until the second brood. We provide empirical evidence supporting Seger's model. Further work, measuring brood sex ratios and comparing abilities of first and second brood males to compete for fertilizations, is required to fully parameterize the model.

**Data Availability Statement:** All data files are available from the Open Access Framework database https://osf.io/k9tvf/?view_only=730772d6e60c4bb0a01cf7ec3f42b9fa.

## Introduction

Eusociality is a complex social system characterised by division of labour between reproductive (queens) and non-reproductive (worker) castes, the latter of which forfeit their own reproduction to help raise genetic relatives, typically siblings [1, 2]. Why a daughter would help raise the offspring of her mother at the cost of her own reproduction has been investigated by some of the most influential evolutionary biologists [3–7]. For helping behaviour to evolve, conditions must be present that make the indirect fitness benefits of helping outweigh the cost of forego-ing personal reproduction that is paid by the helper [6, 8]. Elucidating which physiological, genetic, behavioural and environmental conditions initially favoured helping behaviour and

**Funding:** This work is part of a project that has received funding from the European Research Council (ERC) under the European Horizon's 2020 research and innovation programme (grant agreement No.695744 awarded to JF). The funders had no role in study design, data collection and analysis, decision to publish, or preparation of the manuscript.

**Competing interests:** The authors have declared that no competing interests exist.

set the stage for the evolution of eusociality remains a key challenge in social evolution research [9–12].

Eusociality has evolved independently more times in the Hymenoptera (wasps, bees and ants) than in any other taxa [1]. Eusociality has therefore been intensively researched in hymenopterans, revealing that a whole suite of interacting traits may be required to initiate its evolution. This may explain why, despite the huge ecological success of eusocial Hymenoptera, the majority of hymenopterans are solitary, with each female rearing her offspring alone [13]. Traits that may act as pre-adaptations favouring eusociality include: haplodiploidy, female monogamy, hibernation by mated females, a long period of offspring dependency relative to adult lifespan, and a partially bivoltine life cycle with alternating sex ratios [14, 15].

The unique genetics of haplodiploid organisms in which fertilised eggs become diploid females and unfertilised eggs haploid males may have played a key role [15, 16]. Indeed, as first exemplified by Hamilton's 'haplodiploidy hypothesis', haplodiploid sisters are super-related (as long as they share the same father), such that a female is related to her sister by 0.75 but to her offspring by only 0.50 [4, 5, 17]. A first brood female that emerges while her mother is still alive and reproducing, therefore, stands to gain a greater fitness benefit by helping to raise her second brood sisters than by raising a brood of her own [18, 19]. However, haplodiploidy also results in females being less related to their brothers (r = 0.25) than to their sons, exactly cancelling out the benefit of super-sister relatedness [20, 21]. Sex ratios that are split either temporally (i.e., across broods) or demographically (i.e., female-biased ratios in potential altruists' nests and male-biased in solitary nests) are therefore required for helpers to take advantage of sister super-relatedness [6, 22, 23].

Many models based on hymenopteran life histories have been created to understand the conditions required for eusociality to evolve. One of the best-known, developed by Jon Seger [22], showed how an interaction between genetic, behavioural and ecological traits could favour eusociality in some lineages but inhibit it in others. Two key life history traits are required for eusociality to be favoured in Seger's model. The first is a partially bivoltine life cycle where mated females overwinter, found new nests in spring, then rear two broods of offspring. First brood daughters therefore have the opportunity to help their mothers rear the second brood, if this is favoured by selection. Second, a proportion of first brood males must survive long enough to compete with second brood males for matings with second brood females. This increases the reproductive value of males relative to females in the first brood, because by mating with females of both broods, first brood males produce more offspring than do first brood females [6]. This, in turn, leads foundresses to produce a male-biased first brood sex ratio. Then later in the season, the presence of long-lived first brood males when the second brood reach adulthood reduces the value of second brood males, thus favouring a correspondingly female-biased sex ratio in the second brood. This sets the stage for the evolution of helping: by assisting their mothers to rear a female-biased second brood, first brood daughters can capitalize on super-sister relatedness.

Three key predictions emerged from Seger's model. First, eusociality should arise more often in haplodiploid lineages that over-winter as inseminated females than in lineages that over-winter as unmated adults or larvae. Second, in solitary or incipiently social populations, the first brood should be male-biased and the second, later in the season, female-biased. Corresponding with these predictions, partially bivoltine life cycles and over-wintering by mated females are widespread in temperate hymenopteran lineages that include eusocial species, such as halictine bees and polistine wasps [22]. Studies of hymenopterans with bivoltine life histories have also revealed links between alternating brood sex ratios and the presence and absence of eusociality [24]. A third prediction, however, has been little tested: in solitary or incipiently social populations, some first brood males must survive long enough to mate with

second brood females. If this does not happen, foundresses are no longer predicted to produce a female-biased second brood sex-ratio, so that first brood females cannot capitalize on super-sister relatedness by helping their mothers.

Following Seger [22], models of social evolution have begun to include male survivorship across broods as a key parameter [15]. However, to evaluate such models as foundations for understanding the evolution of eusociality, it is essential to elucidate whether males are indeed able to survive across broods. Seger [22] stated that "it is reasonable to suppose" that hymenopteran males from the first brood survive long enough to mate again in the second brood. Of the four examples Seger cited in support of this statement, however, only two were from systematic, field-based census studies; both of which were on species without mated female hibernation [25, 26]. Seger also stated that males would have to survive at least 4–8 weeks to mate across both broods, but there is little empirical evidence of male longevity to support this assumption. Indeed, male Hymenoptera are widely regarded as "short-lived vessels of sperm" [8]. In order to gauge the efficacy of Seger's model in explaining the evolutionary origins of eusociality it is therefore vital to empirically measure male survivorship in additional species.

Seger's model may be relevant only for investigating which factors may be involved in the initiation of helping behaviour in solitary and incipiently eusocial populations [15]. Once helping by first brood females evolves, helper efficiency may increase so that selection favours foundresses producing all-female first broods, as observed in many obligately eusocial taxa [27]. In the absence of first brood males, the two sexes of the second brood become equally valuable and the second brood sex-ratio should return to 1:1, so that by helping, first brood females can no longer capitalize on super-sister relatedness. There may be caveats to this argument, such as the possibility that first brood males are still produced because they can mate with first brood females that become replacement queens following early foundress death. Evidence for this caveat, however, is inconsistent with first brood males occurring in some populations of obligately eusocial Hymenopterans, but not in others despite frequent early foundress death [J. Field, personal observation; 27]. Nevertheless, the ideal taxa in which to investigate Seger's mechanism are solitary and especially incipiently eusocial populations in which there are two broods, only some first brood daughters become helpers and first brood males are produced.

Here, using capture/mark/recapture (CMR hereafter) techniques we examined male survivorship in just such a species, the facultatively eusocial sweat bee, *Halictus rubicundus*. As assumed by Seger [22], *H. rubicundus* has a partially bivoltine life cycle with over-wintering by mated adult females that found new nests in Spring [9, 22]. Furthermore, *H. rubicundus* exhibits social plasticity in response to environmental and social factors with both social and solitary nests occurring within some populations, while other populations are entirely solitary [28, 29]. Hence, *H. rubicundus* represents a transitional social stage, a potential precursor to highly derived eusocial taxa, making it an excellent species in which to empirically examine which conditions favoured the evolution of eusociality.

## Materials and methods

### Study site

Our study site was native nesting habitat of *Halictus rubicundus* at Trewalla on Bodmin moor, Cornwall, United Kingdom (50°30′45″N; 004°27′59″W). The south facing site, comprised ~3 m of steep, unvegetated sandy bank and a ~2 m flatter patch of mixed grass, bracken and open soil. The surrounding habitat comprised open moorland with gorse as the predominant flowering vegetation. Average annual rainfall and temperature at the site are 1385.2 mm and 13.2° (max): 7° (min) respectively. At the beginning of the study, there were ~74 foundress nests at

the site. To determine nest locations and whether nest occupants became social or solitary (southern native populations can have a mix of social and solitary nests; Field et al. 2010), we observed bee activity at nest sites from near the beginning of the active season (20th May 2019) when over-wintered females become active. We marked all active nests with a numbered nail.

We monitored male bees for 70 days, from their first emergence on 11th July 2019 until the end of the annual activity cycle; ~19th September 2019 (termed the 'study period' hereafter).

Males were caught using an insect net, marked for individual identification, measured and scored for condition. Males were marked by applying different coloured spots of enamel paint to the thorax; size was determined by measuring forewing length from the outer edge of the tegulae to the wing tip using digital calipers. Male condition was scored on a binomial scale using wing darkness/lack of damage, dense thoracic hair and presence of all limbs (e.g., wings and legs) as 1 = good condition, and lighter/'nibbled'/tattered wings, balding thorax or loss of limb/s as 0 = poor condition.

During the 70-day study period, we conducted a census of males from ~0800–1800 on every day when the weather was suitable for bee activity (bees were not active during heavy rain or high winds), totalling 28 days (termed 'census days' hereafter). Census days consisted of one researcher observing the nesting site for male bees emerging from nests, free-flying males and, in second brood, males emerging from nests into traps that had been placed over nest entrances. Meanwhile, another researcher walked a standardised ~500 m diameter circle around the site looking for males that were free-flying or resting on flowers and other vegetation. It was not possible to record data blind because our study involved focal animals in the field. In *H. rubicundus*, as in other temperate sweat bees, the cessation of provisioning by first brood workers is a very clear break in the annual life cycle before second brood reproductives emerge [24; Fig 1]. In our study population, daily behavioural observations were used to determine that first brood workers ceased provisioning on the 28th July 2019 (Fig 1). To ensure that we could identify when second brood gynes and males emerged, we covered 37 nests, where worker provisioning had been observed, with traps following cessation of worker provisioning. We were, therefore, confident that any males caught in the traps following this period were second brood males. During this time, we also continued to capture free-flying males. To get an overall idea of male size, condition and longevity, we included second brood males in our daily census; marking them and recording their size and condition. However, since the main aim of our study was to test whether first brood males survive until the second brood, we used data only from first brood males in our analyses. To further ensure that only first brood males were included in our analyses, we used census data only from males that were captured at least one week before the transition period from the cessation of first brood worker activity and second brood reproductive emergence (29/7/2019). The first second brood female emerged on 2nd August 2019; the first second brood male was captured on the 3rd August 2019. In total, we captured 54 first brood and 32 second brood males, and 13 in the transition period between the first and second broods.

## Data analysis

We used the program 'MARK' [30] to estimate apparent survival probabilities of male *H. rubicundus*. Here, we use the term 'apparent survival' since it was not clear whether an individual that was not recaptured had perished or dispersed from the sample site [31, 32]. We used the 'RMark' package [33] in the program R (R core team 2020–21) as a formula-driven interface to create Cormack-Jolly-Seber [CJS hereafter, 34–36] survival and capture estimation models using the program 'MARK'. Cormack-Jolly-Seber models estimate two parameters for individual capture histories. The first parameter is survival (Phi), the probability that individuals alive

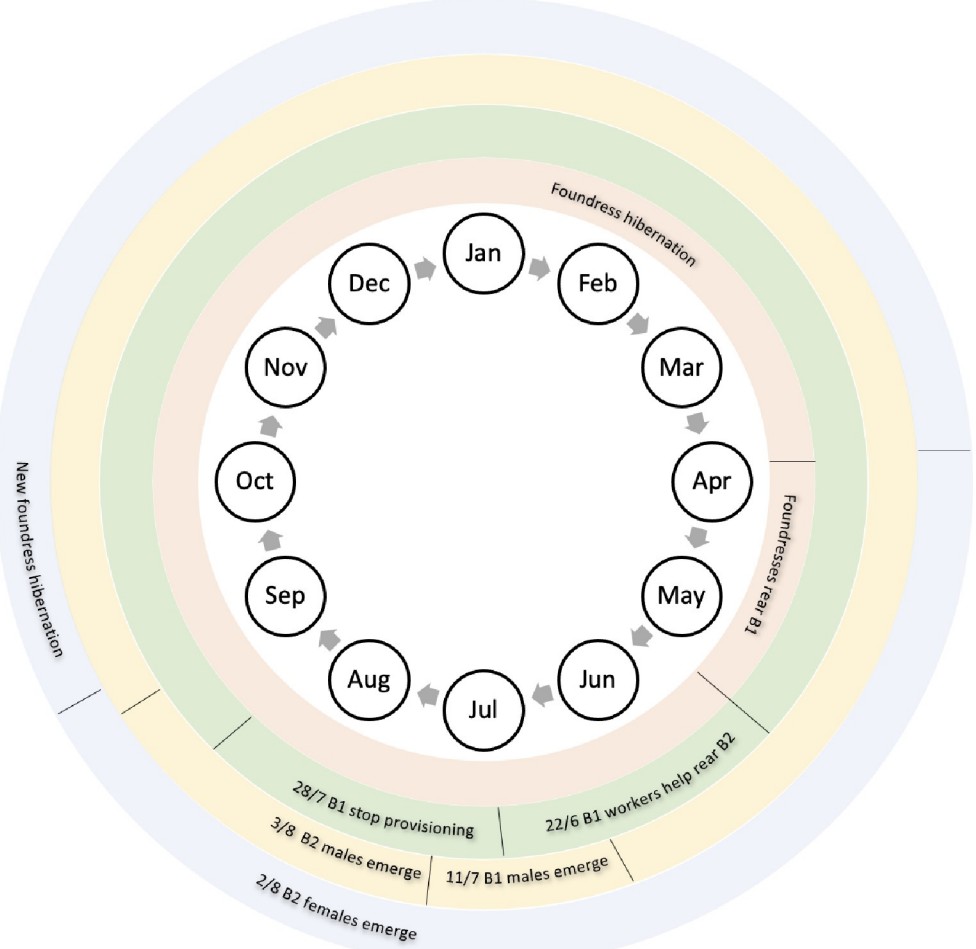

**Fig 1. Key points in the annual lifecycle of original foundresses (orange), first brood worker females (green), first brood and second brood males (yellow), and second brood reproductive females/new foundresses (blue) of the facultatively eusocial sweat bee _Halictus rubicundus_.**

at one capture occasion survive until the next. In this study the individual sampling periods were days and hence the survival estimate would be the probability that an individual caught on one sampling day survives until the next. The second parameter, recapture (p), estimates the probability that marked individuals alive on a capture occasion (in this study a single capture day) are actually re-captured [37]. Furthermore, CJS models allow the user to test the significance of variables that may affect these parameters such as age, size and condition [38].

We created CJS models which varied in whether survival and recapture parameters were held constant (an intercept-only model; e.g., Phi[~1] p[~1]) or varied as a function of different biological predictor variables namely, size (a continuous variable = ~size) and condition (a two-factor categorical variable comprising 'good' and 'poor' condition = ~condition; Table 1). In CJS models, 'time' (e.g., in this study, the day along the 28-day census period when the individual was initially captured—e.g., day two) and 'age' (longevity; that is, the time from initial capture–e.g., 4 days) are default variables. The most appropriate model for making inferences about male survival was selected using Akaike's information criterion [AICc hereafter; 39], weights and deviances.

**Table 1. The top five formulations of the Cormack-Jolly-Seber model (based on Akaike's information criterion, AICc; [39]) used for testing the effect of size (wing length in mm) and condition (a two-factor categorical variable comprising 'good' and 'poor' condition) on estimates of male *H. rubicundus* survival across first brood and second brood during a ~70-day capture/mark/recapture study.** The model with the lowest AICc was deemed the best fit for the data. Phi = survival; p = recapture [31].

| Model | Npar | AICc |
|---|---|---|
| **Phi(~1) p(~1)** | **2** | **225.14** |
| Phi(~condition) p(~condition) | 4 | 226.11 |
| Phi(~1) p(~condition) | 3 | 226.44 |
| Phi(~condition) p(~1) | 3 | 226.77 |
| Phi(~1) p(~size) | 3 | 227.03 |

The model with the lowest AICc is highlighted in bold text.

## Results

During the 28 census days (i.e., days when the weather was suitable for bee activity, and hence for conducting a census) of the 70-day study period (i.e., all sequential calendar days over which the study was conducted), a total of 99 males were captured, marked and measured; 54 first brood, 32 second brood, and 13 males that were caught in the transition period between the first and second brood. Of these individuals, 33 were recaptured (22 first brood; 11 second brood–two from beneath nest traps) and hence, ~66% of marked males were captured only once. Most males were in good condition at first capture (e.g., only five out of 54 first brood males were in poor condition) and tended to maintain their condition throughout the census period. Of first brood males that were scored as being in good condition upon first capture, only three (5%) of those recaptured decreased in condition during the census period. Of all first brood males captured, 48% were larger than the median male size of >6.8mm (wing length). A higher percentage of large males were recaptured more than once (28%) compared with small males (11%). More first brood males were captured for the first time in the early stages of the emergence period (weeks one and two) compared with later on in the census (weeks 3 and 4; Fig 2). Six males from the first brood that were first captured at least 7 consecutive study period days (e.g., sequential calendar days) before the second brood period began (~2/8/2019) survived into the second brood emergence period. The longest interval between initial capture and final recapture of an individual was 22 days of the 70-day study period. The average number of days between first and last capture was 4.94 (Fig 3).

## Survival analysis

Based on AICc values, the model that best fit our first brood male data was the intercept only model (Phi[~1], p[~1]) where survival and recapture probabilities remained constant that is, they did not vary as a function of time or covariates (Table 1). A second model in which survival and recapture probabilities were allowed to vary as a function of condition (Phi[~condition], p[~condition]) is also worth consideration since it was only a few AICc points of separation from the best model (Delta AICc between the two models was 1.2: Table 1). The highest-ranking model estimated the probability of male survival from one sampling occasion to the next as 0.80 (±0.23 SE). The second highest-ranking model (Phi[~condition], p[~condition]), estimated the survival probability of males in good condition as 0.81 (±0.24 SE) and males in poor condition as 0.53 (±0.79 SE). Males in good condition, therefore, had a 28% greater apparent survival compared with males in poor condition. Furthermore, estimated average lifespan of males in good condition was 4.74 census days while for males in poor

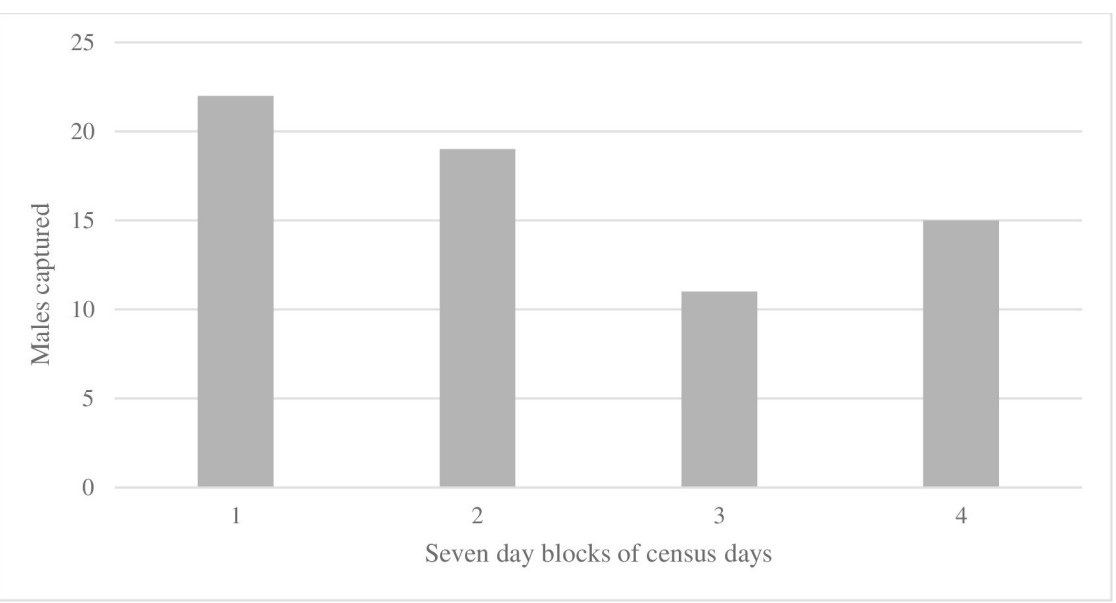

**Fig 2. The number of *Halictus rubicundus* males captured for the first time over 28 census days (i.e., days with suitable whether or sampling bees) in seven day blocks during a 70-day (e.g., 70 consecutive calendar days) capture/mark/recapture study.**

condition it was 66% lower at only 1.58 census days. The relatively low average lifespan estimates are likely to be due to the high number of males recorded only once during the census.

The highest-ranking model estimated the probability of recapture on any given sampling occasion as 0.14 (±0.27 SE). The second highest-ranking model (Phi[~condition], p

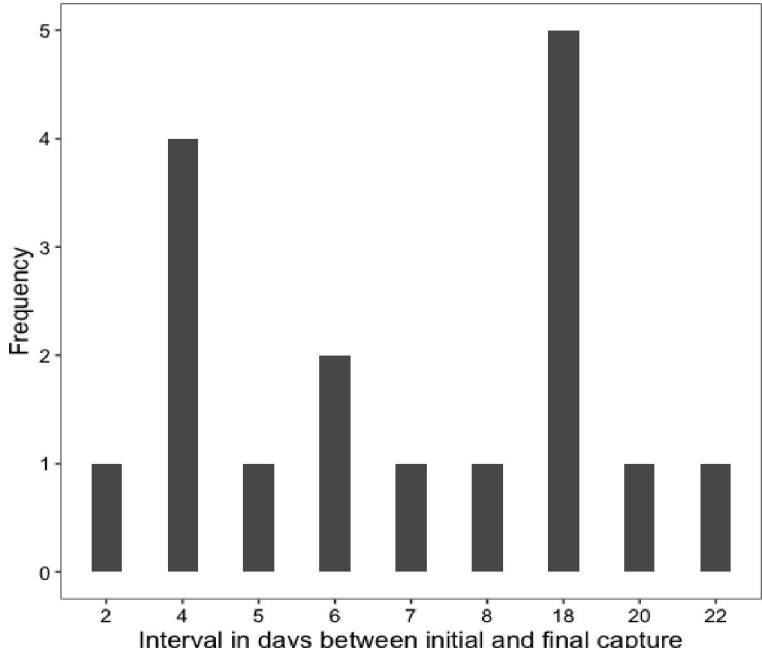

**Fig 3. The interval in days between initial and final capture of male *Halictus rubicundus* during a 70-day capture/mark/recapture census.**

[~condition]) estimated the recapture probability of males in good condition as 0.13 (±0.29 SE) and males in poor condition as 0.52 (±1.27). In the second highest-ranking model, the relatively high standard errors around estimates for survival and recapture probabilities of males in poor condition may be due to the low sample size of males in 'poor' condition (five males were in poor condition; 49 males were in good condition). Similar issues arose in the estimates produced by the other top four-ranking models and hence, we deemed the highest-ranking, intercept-only model as the 'best' fit for our data (Table 1).

## Discussion

Of the 54 *H. rubicundus* males that were captured and marked in the first brood, six (11.1%) were recaptured in the second brood, providing support for Seger's prediction that first brood males are able to survive across broods. Furthermore, the longest recorded survival of a male in our census was 22 days of the entire 70-day study; a timeframe that easily extends across the boundary between the first and second brood (~6 days) in this species. In our analyses, the highest-ranking model held survival and recapture probabilities constant that is, they did not vary as a function of the covariates size and condition (Table 1). Models which included size and condition not only had lower AICc values (Table 1) than the intercept-only model but, for the models including condition, there were also high standard errors around survival and recapture estimates for males in poor condition, suggesting that the small sample size of males in poor condition rendered the model estimates unreliable. Male survival probability as estimated from the highest-ranking CJS model was high (0.80 ±0.23) lending further support for Seger's assumption of male longevity. In contrast, the estimate for recapture probability was low (0.14, ±0.27).

### Male dispersal

Given the observed survival of six first brood males across broods, and the high survival probability estimates of the CJS model, it seems reasonable to suppose that more first brood males than were recaptured survived until the second brood. A low recapture rate may be due to dispersal of males from their natal territory. Indeed, while little is known about male Hymenopteran dispersal, it seems that male dispersal and female philopatry may be widespread in this group [40, 41]. Of our entire census population (that is, first brood, second brood, and males captured during the transition period), 66 males were captured and marked only once suggesting that dispersal may be common in this species.

Several factors are likely to affect male hymenopteran dispersal behaviour which, in turn, would affect the accuracy of survival estimates. For example, the propensity of males to disperse from their natal territory may vary within a population due to different mating strategies and factors such as nest density, female mating mode (e.g., monoandry/polyandry), male conspecific density, patchiness of nests and resource areas, and body size. Alternative mating strategies are known to be related to size in some hymenopterans; larger males remain around nests to guard and monopolise newly-emerged females, while smaller males disperse to mate with foraging females [42]. However, we found no links between male size and recapture probability during the census period. Observations of *Halictus rubicundus* populations in New York suggested that, while females tend to be philopatric, males have a mixed dispersal/mating strategy with some commonly returning to natal and non-natal nest-sites while others actively 'patrol' vegetation around nest sites, searching for females [43].

Female mating mode can also strongly influence a male's decision to disperse or remain in the natal territory. For example, if females mate only once then males that remain near nests may gain a significant fitness advantage by being the first to mate with emerging, virgin

females. Contrastingly, if females mate multiply then there may be less of an advantage to being the first to mate with a female and hence, dispersing to search for mates on flowers may increase the number of mates encountered [44]. Generally, female halictine bees are known to be able to physically resist mating attempts [45, 46]. Very little is known about female mating mode in *H. rubicundus*, but a mix of strategies may be employed, with both single and multiple mating occurring even within populations (offspring genotyping revealed that 3/15 nests contained offspring from multiple mating events; Rebecca Boulton *pers com*).

Alternatively, the low number of recaptures in this study may be due to high male mortality. Indeed, many abiotic and biotic factors such as weather, food availability, predators and parasites may affect male survival and it may be that mortality rates in male sweat bees are high [47]. To further elucidate the actual survival rates of male *H. rubicundus*, investigation into the effects of dispersal-related factors such as female mating mode and male mate-searching/mating, and the effects of abiotic and biotic factors on both dispersal and male survival would be fruitful.

Seger [22] stated that males need only survive for 4–8 weeks to mate with females across broods, and selection could favour high survivorship because mating with two sets of females would greatly increase male fitness. In *H. rubicundus*, there is a clearly observable divide between the cessation of worker activity in the first brood and the emergence of fresh, second brood gynes (Fig 1). At our field site, this transition occurred over an ~6-day period. A late-emerging first brood male would, therefore, have to survive only ~6 days in order to mate with females from both broods. Furthermore, the longest recorded survival of a male in our census was 22 days; a timeframe that easily extends across the boundary between the first and second brood. Of course, the earlier first brood males emerge and the longer they can survive into the second brood, the greater their opportunities to mate across both broods. There is, however, likely to be an optimal time of first brood male emergence based around the complex interaction of many factors such as time of female emergence, aging, male mating competency and female mating strategy. Furthermore, in *Halictus rubicundus*, other factors such as temperature (more male eggs tend to be produced early in the brood one stage in warmer years) and foundress control of offspring sex ratios may affect how early in the brood one period males are produced [43, 48]. For example, social nest foundresses may benefit from a female-biased sex ratio early in brood one since the rarity of males would increase the likelihood that emerging daughters would remain unmated: unmatedness is a possible cue causing offspring to stay and help provision the foundresses' second brood [48, 49 but see; 50].

## Male competitive ability

By covering nests with traps, we were able to observe that second brood females began to emerge at about the same time (one day before) second brood males. This may provide an advantage to first brood males surviving to the second brood as they would have first access to freshly emerged, unmated females. Furthermore, first brood males may have the advantage of mating experience, including mate-searching and competition for matings, unlike freshly emerged second brood males [51]. A counterargument could, of course, be that first brood males would be at a disadvantage compared with freshly emerged second brood males due to decreases in vitality and fecundity with age [52].

Indeed, when considering the fitness advantages of first brood males surviving until the second brood, there is likely to be a complex trade-off between their time of emergence, previous mating experience, and decreases in vitality, fecundity and competitive ability. For example, even if males are able to survive across both broods, it is vital that they are still able to mate in order for Seger's predicted effect of male survival on alternating sex ratios and eusociality to be

valid. First brood males may no longer be reproductively viable in the second brood for several reasons: 1. males have a finite amount of sperm which may be depleted by the second brood; 2. the quality of sperm may be reduced making them non-viable later in the season, and 3. older males from the first brood may be unable to compete for matings against younger, second brood males [53, 54]. Future studies examining male mating competency and sperm quality/competition among males from both broods would provide vital insights. To further investigate the role of first brood male survival in the evolution of eusociality, it is crucial to test if a male-biased sex ratio and long-lived males in first brood do indeed confer a higher fitness advantage to foundresses, thus playing a key role in the evolution of alternating brood sex ratios and eusociality.

## Conclusion

Here, we have shown that first brood males of a Hymenopteran species with a partially bivoltine lifecycle are able to survive into the second brood, lending support to a key prediction of Seger's [22] model and its efficacy in predicting which pre-adaptations may be required to trigger the evolution of eusociality. To more fully parameterize Seger's model we would need to establish the extent to which *H. rubicundus* meets his other key predictions. For example, we know that *H. rubicundus* has a partially bivoltine life cycle with over-wintering by mated adult females that found new nests in Spring, and that social foundresses produce two broods per annual cycle [29, 55]. The majority of over-wintering females are thought to be the foundresses' brood two offspring. However, some mated brood one females produce offspring in the same year which then go on to mate in brood two and subsequently over-winter as mated adults, but further investigation is needed [i.e., a bivoltine lifecycle; 29]. Another important step would be to investigate whether Seger's prediction of a male-biased sex ratio in the first brood and a female-biased sex ratio in the second brood occur in nature. As well as male survival, factors such as within-population split sex ratios [6, 56, 57], facultative maternal investment, over-wintering offspring decisions or the sex of any offspring produced by workers may lead to alternating brood sex ratios [23, 24, 56]. It would, therefore, be fascinating to use the facultatively eusocial *H. rubicundus* in which solitary and social nests can occur within the same population, or in different populations [28, 29], to investigate alternating sex ratios close to the evolutionary origin of eusociality.

## Acknowledgments

We would like to thank Charlie Savill and Lewis Flintham for their assistance with field work. We thank anonymous reviewers for their useful comments on this manuscript.

## Author Contributions

**Conceptualization:** Jodie Gruber, Jeremy Field.

**Data curation:** Jodie Gruber.

**Formal analysis:** Jodie Gruber.

**Funding acquisition:** Jeremy Field.

**Investigation:** Jodie Gruber.

**Methodology:** Jodie Gruber.

**Project administration:** Jodie Gruber.

**Visualization:** Jodie Gruber.

**Writing – original draft:** Jodie Gruber.

**Writing – review & editing:** Jodie Gruber, Jeremy Field.

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
