## [Decision Letter · Decision Letter 0]

26 May 2022

PONE-D-22-06259Male survivorship and the evolution of eusociality in partially bivoltine sweat bees.PLOS ONE

Dear Dr. Gruber,

Thank you for submitting your manuscript to PLOS ONE. After careful consideration, we feel that it has merit but does not fully meet PLOS ONE’s publication criteria as it currently stands. Therefore, we invite you to submit a revised version of the manuscript that addresses the points raised during the review process.

All three referees and I think that your MS is much improved and will be ready fo acceptance provided you give an extra attention to the minor comments made. In particular, table 1 seems to be missing or incomplete.

We look forward to receiving your revised manuscript.

Kind regards,

Nicolas Chaline

Academic Editor

PLOS ONE

Journal Requirements:

2. Thank you for stating the following in the Acknowledgments/Funding Section of your manuscript: 

This work is part of a project that has received funding from the European Research Council (ERC) under the European Horizon’s 2020 research and innovation programme (grant agreement No.695744 awarded to JF). 

This work is part of a project that has received funding from the European Research Council (ERC) under the European Horizon’s 2020 research and innovation programme (grant agreement No.695744 awarded to JF). The funders had no role in study design, data collection and analysis, decision to publish, or preparation of the manuscript.

5. Please include your tables as part of your main manuscript and remove the individual files. Please note that supplementary tables (should remain/ be uploaded) as separate "supporting information" files.

Reviewers' comments:

Reviewer's Responses to Questions

**Comments to the Author**

1. Is the manuscript technically sound, and do the data support the conclusions?

Reviewer #1: Yes

Reviewer #2: Yes

Reviewer #3: Yes

2. Has the statistical analysis been performed appropriately and rigorously? 

Reviewer #1: Yes

Reviewer #2: Yes

Reviewer #3: I Don't Know

3. Have the authors made all data underlying the findings in their manuscript fully available?

Reviewer #1: Yes

Reviewer #2: Yes

Reviewer #3: Yes

4. Is the manuscript presented in an intelligible fashion and written in standard English?

Reviewer #1: Yes

Reviewer #2: Yes

Reviewer #3: Yes

5. Review Comments to the Author

Reviewer #1: The paper investigates the occurrence of male survival across the two broods of the partially bivoltine sweat bee Halictus rubicundus. Determining the presence and degree of male survival in a partially bivoltine species is crucial for models for the evolution of eusociality. These models – first proposed by Seger – suggest that male survival of first brood males can lead to a male-biased first and female-biased second brood. In haplodiploid species, this causes females from the first brood to be more closely related to their mother’s offspring from the second brood than to their own offspring. As a consequence, helping behaviours are favoured by natural selection, even if benefit-cost ratios of helping (in terms of Hamilton’s rule) are slightly below 1.

I found the paper to be important to test how plausible some of the assumptions of these models are. The main finding – that male survival occurs across broods – is convincing, even without any statistical analysis, because the authors recaptured first brood males during the second brood period and present data on male longevity. The statistical analysis seems to be thoroughly done, and the authors are careful to highlight possible limitations, e.g. small sample sizes of recaptured males in poor conditions.

My overall assessment is therefore very positive. However, unfortunately, I could not find Table 1 in my version of the manuscript. Here are some further minor comments:

Line 80-82: I find the expression “some of the greatest minds in evolutionary biology” to be unverifiable. Maybe replacing it by “most influential evolutionary biologists” would be better, but this is still a subjective claim. I am not sure if it is necessary at all that the reader knows who else previously worked on these questions.

Line 92: Looking at the Dew et al. 2018 paper, I am not sure if it really supports the claim made here.

Line 165-166: It is not so clear which aspect of Seger’s model “The mechanism” refers to. Also, strictly speaking the mechanisms that initiates helping is mutation. It is then details of Seger’s model, i.e. a partially bivoltine life cycle, haplodiplody, female hibernation, and so on, that affect whether helping is favoured by selection. None of these things initiates helping.

Line 189-190: I fully understand that the authors present their study species as “a transitional social stage, a potential precursor to highly derived eusocial taxa”. As this is an argument very commonly made and accepted by many, the authors may feel free to ignore my comment. However, it is not clear to me why a social system as that of Halictus rubicundus should exist at all if it was a transitional stage for the evolution of eusociality. If sociality, as in Halictus rubicundus, should be maintained and not to be driven to advanced eusociality, something has to prevent that from happening, e.g. absence of lifetime monogamy or whatever.

Line 349: I think the contrast here should rather be monandry vs polyandry since the authors are talking about female mating mode. In Seger model, males of course need to be able to mate multiple times, contrasting monogamy.

Line 369-370: I found this sentence a bit confusing. It must be true that dispersal relates to recapture probability, since more dispersal means less recapturing, if you always sample in the same area. It is also not entirely clear to me how this sentence relates to the discussion of mating frequency above it. Could this maybe be rephrased to be clearer?

Reviewer #2: The manuscript of Gruber and Field present field estimates of a key life-history parameter that theory predicts to have major consequences in the evolution of eusociality in haplodiploid organisms. The conceptual justification of the study is flawless. The authors clearly have a deep understanding of the theoretical models that predict male survival to be key in the evolution of eusociality. In line with that understanding, they have chosen a species that, due to its ecology, life-history, social organization and evolutionary origins, can shed light into the viability of an evolutionary path towards eusociality. The methods use are shockingly simple and fitting to the question at hand. I highly appreciate that using such straightforward methodology and clear conceptual justification the authors bring forward our understanding of the life-history of the species and contribute to the understanding of a major evolutionary transition.

There is only one problem that prevents me from directly and fully endorsing the publication of this manuscript. I could not find the Table 1 to which the authors refer to for the results of the statistical analysis. This is probably a mistake in the submission process. However, it is my duty to evaluate whether the reported results are justified in the statistical inference. Thus, I recommend the authors to make their statistical analysis available, not only the table but the code as well. This is in line with the aim of having a more open and transparent scientific process.

Just one minor comment, in lines 110 to 114 there is a small inaccuracy. Authors state that “super-relatedness” of haplodiploid sisters under even sex ratios is not enough to promote the evolution of help. This is however not necessarily correct. If relatedness is measure as the so-called life-for-life relatedness, which is a measure of both relatedness and reproductive values, then super-relatedness under even sex-ratio can promote the evolution of helping. In other words even if sex ratios are even, as long as females have a higher reproductive value, selection can favour the evolution of helping. This is shown in Quiñones and Pen (2017), male survival together with female hibernation and haplodiploidy promote the evolution of helping even if foundresses can´t flexibly adjust brood sex ratios and instead produce even broods in both reproductive events. This is not a major issue obviously, but it is worth setting straight.

Reviewer #3: Main comments

I read a previous version of this paper. It seems to me that this version is better focussed on the main issue, which is testing one aspect of Seger’s 1983 hypothesis that partially bivoltine life cycles may promote the evolution of brood sex ratios that might then favour the evolution of eusociality. The extent to which Brood 1 (G1) and Brood 2 (G2) sex ratios differ, should depend on several factors, including survival rates and lifespans of males. This is because male lifespan dictates whether brood 1 males live long enough to compete with brood 2 males for mating opportunities with brood 2 females. This paper uses mark-recapture methods to investigate survival rates and longevity of brood 1 males of Halictus rubicundus, a facultatively eusocial halictine that in many locations, exhibits partial bivoltinism. It is therefore an excellent species for testing Seger’s hypothesis. Mark-recapture is an interesting and challenging approach for measuring survival in tiny insects, especially in the sex that likely disperses. However, it seems to have been fairly successful here, demonstrating that a proportion of Brood 1 males do indeed survive long enough to encounter, and presumably mate with, females of Brood 2.

This is a worthwhile result, but I do suggest that a few revisions would improve the paper.

Introduction

The Introduction, especially lines 77-117, currently restates the textbook background for why the evolution of eusociality is fascinating. However, almost 60 years after Hamilton proposed kin selection as the solution to the altruism “paradox”, and lots of empirical and theoretical studies confirming that kin selection works, it really isn’t paradoxical anymore. Also, this paper is not about relatedness, so lines 102-117 could be deleted. Deleting the classical, textbook summary would allow the Intro to proceed to the issue of sex ratios and Seger’s model, which currently does not get mentioned until line 120.

Line 94. “This may explain why”… refers back to the previous two sentences, and so seems to say most Hymenoptera are solitary because multiple traits are involved in social evolution. Is this what you meant?

Lines 130-149. Seger’s models do predict a male-biased Brood 1 sex ratio (M2) and a female-biased Brood 2 sex ratio (M1) – for solitary bivoltine insects. This contrasts with the observed sex ratios of partially bivoltine bees like H. rubicundus, that have female-biased Brood 1 sex ratios. So the Brood 2 sex ratio might favour helping by the few Brood 1 females produced? So now we do have a paradox, because the sex ratio for the first brood is not what we see in bivoltine halictids, especially eusocial ones, which produce female-biased first broods. The next paragraph addresses the predictions of Seger’s models and how they predict eusociality – which results in female-biased broods. I think it would be useful to explain why the lifespan of Brood 1 males is still expected to affect the B2 sex ratio in a bee like H. rubicundus, that has already evolved eusociality, including a female-biased B1 sex ratio.

Methods – Unfortunately, Table 1 was missing from the manuscript, as far as I could tell. It is therefore difficult to assess whether the parameters of the model were sufficiently clearly explained.

Line 212 Tegula – plural tegulae. Not tegulum.

Line 213. Please specify which limbs – legs? wings?

Line 227. What is the Brockmann and Grafen reference for here?

Line 239. I did not understand what the “key first brood/two transition period” was.

Line 261-269. I did not understand what the ‘=’ signs were for. Are they mathematical signs or a short form for English words? Couldn’t check with Table 1, which is missing.

Line 289-290. Does this mean that males were recaptured about 5 days after their first capture (5 days by the calendar) or 5 observation days later? Throughout this paragraph I was a bit confused about which of these intervals was being used. Similarly, Figure 2 refers to weeks, but weeks are not simply collections of 7 days – they are a calendar interval.

Results

Line 283. Statistical evidence or data to support the claim that body size was similar for Brood 1 and Brood 2 males?

Discussion

Line 356: Kukuk did not study H. rubicundus, so this reference seems misplaced.

Line 369. Who is Boulton?

Lines 422. As far as I know, there is no evidence for a male-biased Brood 1 sex ratio in bivoltine, solitary halictids. Check Plateaux-Quenu 1989 who found female-biased B1 sex ratios in L. villosulum, and Stockhammer 1966 for similar pattern in Augochlora pura. At least one other facultatively social species, Megalopta genalis, has been studied extensively and would be a good comparator to H. rubicundus for that reason – moreover, sex ratios have been carefully studied by Adam Smith et al. which seems highly relevant to the current study.

6. PLOS authors have the option to publish the peer review history of their article (what does this mean?). If published, this will include your full peer review and any attached files.

Reviewer #1: No

Reviewer #2: **Yes: **Andrés E. Quinones

Reviewer #3: No

---

## [Author Response · Author response to Decision Letter 0]

9 Aug 2022

Responses to Reviewers' Comments:

• Below, we provide original comments from each reviewer (italics) followed by our response (bold text).

Reviewer #1: The paper investigates the occurrence of male survival across the two broods of the partially bivoltine sweat bee Halictus rubicundus. Determining the presence and degree of male survival in a partially bivoltine species is crucial for models for the evolution of eusociality. These models – first proposed by Seger – suggest that male survival of first brood males can lead to a male-biased first and female-biased second brood. In haplodiploid species, this causes females from the first brood to be more closely related to their mother’s offspring from the second brood than to their own offspring. As a consequence, helping behaviours are favoured by natural selection, even if benefit-cost ratios of helping (in terms of Hamilton’s rule) are slightly below 1.

I found the paper to be important to test how plausible some of the assumptions of these models are. The main finding – that male survival occurs across broods – is convincing, even without any statistical analysis, because the authors recaptured first brood males during the second brood period and present data on male longevity. The statistical analysis seems to be thoroughly done, and the authors are careful to highlight possible limitations, e.g. small sample sizes of recaptured males in poor conditions.

My overall assessment is therefore very positive. However, unfortunately, I could not find Table 1 in my version of the manuscript. 

Response: We thank Reviewer 1 for their comments. We have double-checked the original file submission and the table was uploaded as a separate file as opposed to part of the main body of the manuscript. We have now included the table in the main body of the MS, which we understand is the preferred PLOS One formatting for tables. Having looked at the PDF version originally submitted for reviewers, we discovered that Table 1 was available to be viewed via a link on the page after Figure 3.

Here are some further minor comments:

Line 80-82: I find the expression “some of the greatest minds in evolutionary biology” to be unverifiable. Maybe replacing it by “most influential evolutionary biologists” would be better, but this is still a subjective claim. I am not sure if it is necessary at all that the reader knows who else previously worked on these questions.

Response: As suggested, we have changed the text in line 80-82 from “some of the greatest minds in evolutionary biology” to “most influential evolutionary biologists”. We feel that mentioning who has previously worked on these questions provides background and context that are important to understanding the present field and importance of the research question.

Line 92: Looking at the Dew et al. 2018 paper, I am not sure if it really supports the claim made here.

Response: The Dew et al. 2018 citation has been removed from this section of the MS.

Line 165-166: It is not so clear which aspect of Seger’s model “The mechanism” refers to. Also, strictly speaking the mechanisms that initiates helping is mutation. It is then details of Seger’s model, i.e. a partially bivoltine life cycle, haplodiplody, female hibernation, and so on, that affect whether helping is favoured by selection. None of these things initiates helping.

Response: We have removed the word “mechanism” and changed the text to make it clearer that it is Seger’s model that may only be useful for investigating which factors may be involved in the initiation of eusociality in solitary and incipiently eusocial populations. The new text reads as follows: “ Seger’s (1983) model may be relevant only for investigating which factors may be involved in the initiation of helping behaviour in solitary and incipiently eusocial populations”

Line 189-190: I fully understand that the authors present their study species as “a transitional social stage, a potential precursor to highly derived eusocial taxa”. As this is an argument very commonly made and accepted by many, the authors may feel free to ignore my comment. However, it is not clear to me why a social system as that of Halictus rubicundus should exist at all if it was a transitional stage for the evolution of eusociality. If sociality, as in Halictus rubicundus, should be maintained and not to be driven to advanced eusociality, something has to prevent that from happening, e.g. absence of lifetime monogamy or whatever.

Response: We appreciate the reviewer’s comment regarding transitional social stages and it is an interesting topic for discussion. Agreed, in H. rubicundus, there may be abiotic/biotic constraints on this species transitioning from facultatively eusocial to highly eusocial. However, in our study, it was not a potential social transition in H. rubicundus per se that interested us. Instead, we used H. rubicundus as it represents a potentially similar biology/life history to the past transitional stage of a species that may have gone on to become highly eusocial. Of course, we cannot go into the past to examine the transitional states of now highly derived taxa (such as honeybees). However, we can make predictions about the life histories of those transitional states and find ‘modern’ taxa with similar life histories to enable us to test which factors may drive the evolution of eusociality. Hence, due to its ‘primitively’ eusocial state H. rubicundus acts as an extant ‘proxy’ of past transitional states. No adjustment has been made to the MS in response to this discussion point.

Line 349: I think the contrast here should rather be monandry vs polyandry since the authors are talking about female mating mode. In Seger model, males of course need to be able to mate multiple times, contrasting monogamy.

Response: As suggested, we have changed the text from “monogamy” to “monandry”.

Line 369-370: I found this sentence a bit confusing. It must be true that dispersal relates to recapture probability, since more dispersal means less recapturing, if you always sample in the same area. It is also not entirely clear to me how this sentence relates to the discussion of mating frequency above it. Could this maybe be rephrased to be clearer?

Response: We thank the reviewer for this comment. The agree that the text was unclear and out of context in its position on lines 369-370. Hence, we have moved the sentence to lines 357-358 where it has a relevant link to a discussion about dispersal and size. We have also removed the part of the sentence about dispersal being unlikely to affect recapture since, as mentioned by Reviewer 1, that is clearly not the case.

Reviewer #2: The manuscript of Gruber and Field present field estimates of a key life-history parameter that theory predicts to have major consequences in the evolution of eusociality in haplodiploid organisms. The conceptual justification of the study is flawless. The authors clearly have a deep understanding of the theoretical models that predict male survival to be key in the evolution of eusociality. In line with that understanding, they have chosen a species that, due to its ecology, life-history, social organization and evolutionary origins, can shed light into the viability of an evolutionary path towards eusociality. The methods use are shockingly simple and fitting to the question at hand. I highly appreciate that using such straightforward methodology and clear conceptual justification the authors bring forward our understanding of the life-history of the species and contribute to the understanding of a major evolutionary transition.

There is only one problem that prevents me from directly and fully endorsing the publication of this manuscript. I could not find the Table 1 to which the authors refer to for the results of the statistical analysis. This is probably a mistake in the submission process. However, it is my duty to evaluate whether the reported results are justified in the statistical inference. Thus, I recommend the authors to make their statistical analysis available, not only the table but the code as well. This is in line with the aim of having a more open and transparent scientific process.

Response: We thank Reviewer 2 for their comments and endorsement of our manuscript. While we did include Table 1 in the original manuscript submission, we did not include the table in the main body of the text. Instead, we attached Table 1 as a separate document (as we did for the figures) which resulted in the table only being available as a link in the PDF manuscript after Figure 3. To avoid this confusion, we have now included the table in the main text of the manuscript. As suggested by Reviewer 2, we will also make our code available upon request.

Just one minor comment, in lines 110 to 114 there is a small inaccuracy. Authors state that “super-relatedness” of haplodiploid sisters under even sex ratios is not enough to promote the evolution of help. This is however not necessarily correct. If relatedness is measure as the so-called life-for-life relatedness, which is a measure of both relatedness and reproductive values, then super-relatedness under even sex-ratio can promote the evolution of helping. In other words even if sex ratios are even, as long as females have a higher reproductive value, selection can favour the evolution of helping. This is shown in Quiñones and Pen (2017), male survival together with female hibernation and haplodiploidy promote the evolution of helping even if foundresses can´t flexibly adjust brood sex ratios and instead produce even broods in both reproductive events. This is not a major issue obviously, but it is worth setting straight.

Response: We thank Reviewer 2 for this comment. Reviewer 2’s explanation of how even under an equal sex ratio, helping may still evolve is in accordance with many of the main points in our manuscript. Hence, to avoid misinterpretation, we have removed the ‘even sex ratios’ statement from lines 110-114.

Reviewer #3: Main comments

I read a previous version of this paper. It seems to me that this version is better focussed on the main issue, which is testing one aspect of Seger’s 1983 hypothesis that partially bivoltine life cycles may promote the evolution of brood sex ratios that might then favour the evolution of eusociality. The extent to which Brood 1 (G1) and Brood 2 (G2) sex ratios differ, should depend on several factors, including survival rates and lifespans of males. This is because male lifespan dictates whether brood 1 males live long enough to compete with brood 2 males for mating opportunities with brood 2 females. This paper uses mark-recapture methods to investigate survival rates and longevity of brood 1 males of Halictus rubicundus, a facultatively eusocial halictine that in many locations, exhibits partial bivoltinism. It is therefore an excellent species for testing Seger’s hypothesis. Mark-recapture is an interesting and challenging approach for measuring survival in tiny insects, especially in the sex that likely disperses. However, it seems to have been fairly successfull here, demonstrating that a proportion of Brood 1 males do indeed survive long enough to encounter, and presumably mate with, females of Brood 2.

This is a worthwhile result, but I do suggest that a few revisions would improve the paper.

Introduction

The Introduction, especially lines 77-117, currently restates the textbook background for why the evolution of eusociality is fascinating. However, almost 60 years after Hamilton proposed kin selection as the solution to the altruism “paradox”, and lots of empirical and theoretical studies confirming that kin selection works, it really isn’t paradoxical anymore. Also, this paper is not about relatedness, so lines 102-117 could be deleted. Deleting the classical, textbook summary would allow the Intro to proceed to the issue of sex ratios and Seger’s model, which currently does not get mentioned until line 120.

Line 94. “This may explain why”… refers back to the previous two sentences, and so seems to say most Hymenoptera are solitary because multiple traits are involved in social evolution. Is this what you meant?

Lines 130-149. Seger’s models do predict a male-biased Brood 1 sex ratio (M2) and a female-biased Brood 2 sex ratio (M1) – for solitary bivoltine insects. This contrasts with the observed sex ratios of partially bivoltine bees like H. rubicundus, that have female-biased Brood 1 sex ratios. So the Brood 2 sex ratio might favour helping by the few Brood 1 females produced? So now we do have a paradox, because the sex ratio for the first brood is not what we see in bivoltine halictids, especially eusocial ones, which produce female-biased first broods. The next paragraph addresses the predictions of Seger’s models and how they predict eusociality – which results in female-biased broods. I think it would be useful to explain why the lifespan of Brood 1 males is still expected to affect the B2 sex ratio in a bee like H. rubicundus, that has already evolved eusociality, including a female-biased B1 sex ratio.

Response: We thank Reviewer 3 for their comment and have removed the word “paradox” from line 80. We appreciate Reviewer 3’s point that removing the relatedness explanation from lines 102-117 would result in getting to the discussion of sex ratios and the Seger hypothesis sooner. However, we have chosen not to delete this section. We deem that in order for a broad readership to appreciate the implications of split sex ratios and male survivorship for the evolution of eusociality, and predictions of the Seger model, a brief explanation of the unique relatedness asymmetries in haplodiploids is needed. 

Methods – Unfortunately, Table 1 was missing from the manuscript, as far as I could tell. It is therefore difficult to assess whether the parameters of the model were sufficiently clearly explained.

Response: We included Table 1 in the original manuscript submission, however, we did not include the table in the main body of the text. Instead, we attached Table 1 as a separate document (as we did for the figures) which resulted in the table only being available as a link in the PDF manuscript after Figure 3. To avoid this confusion, we have now included the table in the main text of the manuscript.

Line 212 Tegula – plural tegulae. Not tegulum.

Response: We have replaced “tegulum” with “tegulae”.

Line 213. Please specify which limbs – legs? wings?

Response: We are referring to both legs and wings. We have now included this explanation in the manuscript (L219-220).

Line 227. What is the Brockmann and Grafen reference for here?

Response: The Brockmann and Grafen 1992 reference is cited as a published article that also states there is a the clear break in the annual life cycle of sweat bees.

Line 239. I did not understand what the “key first brood/two transition period” was.

Response: We thank Reviewer 3 for this comment. We have changed the text to make it clearer what we meant by “key first brood/two transition period”. The text now reads “transition period from the cessation of first brood worker activity and second brood reproductive emergence” (L246-247).

Line 261-269. I did not understand what the ‘=’ signs were for. Are they mathematical signs or a short form for English words? Couldn’t check with Table 1, which is missing.

Response: We have removed the “=” symbol to avoid confusion.

Line 289-290. Does this mean that males were recaptured about 5 days after their first capture (5 days by the calendar) or 5 observation days later? Throughout this paragraph I was a bit confused about which of these intervals was being used. Similarly, Figure 2 refers to weeks, but weeks are not simply collections of 7 days – they are a calendar interval.

Response: We thank the reviewer for this comment. Day measures referred to as “the study period” comprised all 70 calendar days of the entire study. Day measures referred to as “census days” comprise only days when the weather was suitable for conducting a census. We have now included this additional explanation (L289-291 & 305). Regarding Figure 2, we agree that the use of the term “weeks” does suggest that the data represents 7 calendar days and not blocks of 7 separate census days. We have removed the term “weeks” from Figure 3 to make it clearer that we are referring to census days.

Results

Line 283. Statistical evidence or data to support the claim that body size was similar for Brood 1 and Brood 2 males?

Response: We thank Reviewer 3 for this comment. We have since removed the statement about B1 and B2 body size from the manuscript since upon reflection this detail is not relevant for the analysis which just included B1 male survival.

Discussion

Line 356: Kukuk did not study H. rubicundus, so this reference seems misplaced.

Reference: This reference has been removed (L 380).

Line 369. Who is Boulton?

Reference: “Boulton” is Rebecca Boulton, a past Postdoctoral Research Fellow from the Field lab who studied H. rubicundus. We have added this extra detail to the manuscript, however, since the reference was an unpublished, personal observation, we have not provided further detail.

Lines 422. As far as I know, there is no evidence for a male-biased Brood 1 sex ratio in bivoltine, solitary halictids. Check Plateaux-Quenu 1989 who found female-biased B1 sex ratios in L. villosulum, and Stockhammer 1966 for similar pattern in Augochlora pura. At least one other facultatively social species, Megalopta genalis, has been studied extensively and would be a good comparator to H. rubicundus for that reason – moreover, sex ratios have been carefully studied by Adam Smith et al. which seems highly relevant to the current study.

Response: We thank Reviewer 3 for their comments and suggestions. We are also not aware of evidence for a male-biased brood sex ratio in bivoltine solitary halictids; it is an interesting area for future research. We are admirers of Adam Smith et al.’s Megalopta work and agree that M. genalis would be a useful species for comparisons with H. rubicundus.

---

## [Decision Letter · Decision Letter 1]

7 Oct 2022

Male survivorship and the evolution of eusociality in partially bivoltine sweat bees.

PONE-D-22-06259R1

Dear Dr. Gruber,

We’re pleased to inform you that your manuscript has been judged scientifically suitable for publication and will be formally accepted for publication once it meets all outstanding technical requirements.

Kind regards,

Nicolas Chaline

Academic Editor

PLOS ONE

Additional Editor Comments (optional):

Reviewers' comments:

Reviewer's Responses to Questions

**Comments to the Author**

1. If the authors have adequately addressed your comments raised in a previous round of review and you feel that this manuscript is now acceptable for publication, you may indicate that here to bypass the “Comments to the Author” section, enter your conflict of interest statement in the “Confidential to Editor” section, and submit your "Accept" recommendation.

Reviewer #1: All comments have been addressed

Reviewer #2: (No Response)

Reviewer #3: All comments have been addressed

2. Is the manuscript technically sound, and do the data support the conclusions?

Reviewer #1: Yes

Reviewer #2: Yes

Reviewer #3: Yes

3. Has the statistical analysis been performed appropriately and rigorously? 

Reviewer #1: Yes

Reviewer #2: Yes

Reviewer #3: Yes

4. Have the authors made all data underlying the findings in their manuscript fully available?

Reviewer #1: Yes

Reviewer #2: Yes

Reviewer #3: Yes

5. Is the manuscript presented in an intelligible fashion and written in standard English?

Reviewer #1: Yes

Reviewer #2: Yes

Reviewer #3: Yes

6. Review Comments to the Author

Reviewer #1: I have reviewed a previous version of this paper. I think that the authors have addressed the comments from the previous round of reviews very well. After carefully reading the manuscript again, I have no further objections and would like to congratulate the authors to their nice work!

Reviewer #2: I appreciate that the authors have addressed all my comments. I think the manuscript is ready for submission. I, however, would like to encourage the authors, to not only make the code available upon request but to share it in a public repository. This, I think should be the new standard in order to make science more open and reproducible. This is a practice we can all profit from and allows others to build upon your work.

Reviewer #3: I think the manuscript is ready for acceptance.

7. PLOS authors have the option to publish the peer review history of their article (what does this mean?). If published, this will include your full peer review and any attached files.

Reviewer #1: No

Reviewer #2: **Yes: **Andrés E. Quiñones

Reviewer #3: No

---

## [Editor Report · Acceptance letter]

11 Oct 2022

PONE-D-22-06259R1 

Male survivorship and the evolution of eusociality in partially bivoltine sweat bees. 

Dear Dr. Gruber:

I'm pleased to inform you that your manuscript has been deemed suitable for publication in PLOS ONE. Congratulations! Your manuscript is now with our production department. 

Kind regards, 

on behalf of

Professor Nicolas Chaline 

Academic Editor

PLOS ONE